# Evolution of the Gram-Negative Antibiotic Resistance Spiral over Time: A Time-Series Analysis

**DOI:** 10.3390/antibiotics10060734

**Published:** 2021-06-17

**Authors:** Hajnalka Tóth, Gyula Buchholcz, Adina Fésüs, Bence Balázs, József Bálint Nagy, László Majoros, Krisztina Szarka, Gábor Kardos

**Affiliations:** 1Department of Medical Microbiology, Faculty of Medicine, University of Debrecen, Egyetem tér 1, H-4032 Debrecen, Hungary; toth.hajnalka13@gmail.com (H.T.); fesus.adina@pharm.unideb.hu (A.F.); balazs.bence@med.unideb.hu (B.B.); nagy.jozsefb93@gmail.com (J.B.N.); major@med.unideb.hu (L.M.); szkrisz@med.unideb.hu (K.S.); 2Doctoral School of Pharmaceutical Sciences, University of Debrecen, Egyetem tér 1, H-4032 Debrecen, Hungary; 3Ostalb Klinikum, Im Kälblesrain 1, D-73430 Aalen, Germany; 4Central Clinical Pharmacy, Clinical Center, University of Debrecen, Egyetem tér 1, H-4032 Debrecen, Hungary; fogyogyszeresz@med.unideb.hu; 5Department of Metagenomics, University of Debrecen, Egyetem tér 1, H-4032 Debrecen, Hungary

**Keywords:** vector autoregressive models, antibiotic consumption, antibiotic resistance, *Escherichia coli*, *Klebsiella* spp., *Pseudomonas aeruginosa*, *Acinetobacter baumannii*, cephalosporin resistance, carbapenem resistance, colistin

## Abstract

We followed up the interplay between antibiotic use and resistance over time in a tertiary-care hospital in Hungary. Dynamic relationships between monthly time-series of antibiotic consumption data (defined daily doses per 100 bed-days) and of incidence densities of Gram-negative bacteria (*Escherichia coli*, *Klebsiella* spp., *Pseudomonas aeruginosa*, and *Acinetobacter baumannii*) resistant to cephalosporins or carbapenems were followed using vector autoregressive models sequentially built of time-series ending in *2015*, 2016, 2017, 2018, and 2019. Relationships with Gram-negative bacteria as a group were fairly stable across years. At species level, association of cephalosporin use and cephalosporin resistance of *E. coli* was shown in 2015–2017, leading to increased carbapenem use in these years. Association of carbapenem use and carbapenem resistance, as well as of carbapenem resistance and colistin use in case of *A. baumannii*, were consistent throughout; associations in case of *Klebsiella* spp. were rarely found; associations in case of *P. aeruginosa* varied highly across years. This highlights the importance of temporal variations in the interplay between changes in selection pressure and occurrence of competing resistant species.

## 1. Introduction

Antibiotic resistance has been a threat to successful anti-infective therapy ever since antibiotics were initially introduced to clinical practice. Emergence of resistance initially inspired discovery of novel drugs active against resistant strains, but, presently, bacteria seem to be at an advantage in this arms race. This is admittedly caused by antibiotic overuse and misuse [1,2,3].

Carlet et al. [4] hypothesized prescriber concern as the most important factor in the choice of antibiotics in empirical treatment. According to this hypothesis, this concern is the driving force for the antibiotic resistance spiral, the process of ever switching towards broader-spectrum drugs when resistance to presently used ones is felt to be threatening, which leads to overuse of and, consequently, resistance to the replacement drug in a vicious cycle until all therapeutic options are exhausted. Appearance of pandrug-resistant isolates among nosocomial Gram-negative bacteria furnishes this hypothesis with an alarming actuality [5,6,7].

This resistance spiral was shown to involve multiple species; emerging cephalosporin resistance in *Escherichia coli* induced increasing preference for carbapenem prescribing, which led to increased carbapenem resistance in *Pseudomonas aeruginosa* and in *Acinetobacter baumannii*. Concern about carbapenem resistance provoked an increased prescribing of colistin as the last-resort drug [8].

This reflects that hospital ecology is characterized by a complex interplay of emergence, ebbing, and re-emerging of various strains of nosocomial pathogens. Pathogen dynamics are strongly interconnected with bacterial evolution; strains may acquire or lose resistance and virulence genes through horizontal gene transfer [9]. New clones of pathogens that develop at or are imported to healthcare institutions compete with the locally dominant members of the microbiota; some new strains are eliminated, some become established, and eventually may become dominant. Alarmingly, in line with the resistance spiral theory, the newly emerging clones are frequently more resistant than former dominant clones [10,11,12,13], thus the volume and pattern of antibiotic consumption frequently have a major impact on local strain dynamics [14].

This work, as an extension of our abovementioned earlier study reporting a snapshot of the resistance spiral [8], attempts to capture the evolution of the resistance spiral over time by examining relationships among drug consumption and resistance and by comparing the interactions between drugs and drug-resistant bacteria at four additional time points.

## 2. Results

The relationship between cumulated resistance and antibiotic consumption was stable across the study years. Cephalosporin consumption was associated with resistance to third generation cephalosporins, cephalosporin resistance was linked to increasing carbapenem use, and increased carbapenem use was associated with carbapenem resistance. Carbapenem resistance was linked to colistin consumption, but this was not related to colistin resistance. These associations were found consistently in all model systems, i.e., in data series ending in 2015, 2016, 2017, 2018, and 2019 (Figure 1, Appendix A). The only relationship showing changes over time was the reciprocal association, i.e., in data series ending in the last four years, not only was carbapenem use associated with increased colistin use, but the increasing colistin use was linked to decreased carbapenem consumption.

This uniformity disappeared when considering species separately (Figure 2, Appendix A). In case of *E. coli*, the link between cephalosporin use and resistance to third generation cephalosporins was consistent across years, except for dataset 2019, but the association between cephalosporin resistance and carbapenem use disappeared in 2017, 2018, and 2019. Curiously, cephalosporin resistance was also associated with increasing cephalosporin use in some years. Testing the resistance spiral further in *E. coli* was precluded by absence of carbapenem resistance in *E. coli* throughout the tested period.

In the case of *Klebsiella* spp. (Figure 2, Appendix A), the link between cephalosporin use and third generation cephalosporin resistance found in series ending in 2015 disappeared and then reappeared in 2019. The association between cephalosporin resistance of *Klebsiella* spp. and increasing carbapenem use was absent in 2015–2017, appeared as a weak link in 2018, then was absent again in 2019. Carbapenem consumption was consistently unlinked to carbapenem resistance of *Klebsiella* spp. in all five datasets. The link between carbapenem resistance of *Klebsiella* spp. and colistin use was present in earlier series, but not in the 2018 or 2019 datasets; in the latter datasets, colistin use was inversely associated with carbapenem resistance.

The relationships in case of *P. aeruginosa* were significant only in the earlier datasets (Figure 2, Appendix A). The only consistent relationship was the absence of association between cephalosporin consumption and ceftazidime resistance of *P. aeruginosa*. The association between ceftazidime resistance of *P. aeruginosa* and carbapenem use, between carbapenem use and carbapenem resistance of *P. aeruginosa*, and between carbapenem resistance and colistin use was only found in the 2015 and 2016 models, but totally absent in other models ( Figure 2; Figure 3, Appendix A). Moreover, in 2015 and 2016 colistin consumption was also associated with decreasing occurrence of carbapenem-resistant *P. aeruginosa* ( Figure 2; Figure 3, Appendix A).

In the case of *A. baumannii*, the associations were highly consistent (Figure 2, Appendix A); the links between carbapenem use and carbapenem resistance, as well as carbapenem resistance and colistin use both were present in all model systems across the tested years.

When following the reconstructed resistance spiral over time, many relationships showed changes across the different years’ models (Figure 4, Appendix A). The initial steps in the tested spiral, the relationships between cephalosporin consumption and cephalosporin resistance of *E. coli* or of *Klebsiella* spp. were never present in the composite models, regardless of the dataset. The next step, i.e., the link between cephalosporin resistance and carbapenem use in the case of *E. coli* was found in datasets 2015 to 2017, but not in datasets 2018 or 2019. Carbapenem resistance was consistently independent of carbapenem use in the case of *P. aeruginosa*, while in the case of *A. baumannii* carbapenem use was associated with increasing carbapenem resistance in 2017 to 2019 (but not in earlier years). A reciprocal effect, i.e., association of carbapenem resistance of *A. baumannii* with decreased carbapenem use, was found in the 2016 and 2017 datasets, but not in the other models. Association of carbapenem resistance with colistin use was variable in the case of *P. aeruginosa*, i.e., the association was found in 2016, but not in other years. However, a reciprocal effect was detected throughout all models, except 2019. In contrast, the association between carbapenem resistance and colistin use was consistently present and unidirectional in the case of *A. baumannii*.

## 3. Discussion

The hospital environment provides a finite number of niches that hospital bacteria may occupy and, consequently, bacteria compete for these. As a result, hospital ecology is frequently characterized by emergence, receding, and re-emergence of different nosocomial strains, creating a complicated interplay of strain dynamics [15]. In this highly competitive environment, any selective advantage may become a major actor in shaping hospital ecology [9,16].

The observation that a shift towards more resistant strains is common in hospitals indicates that antibiotic consumption is a major selective force in hospital ecology and, therefore, is a major driver of strain dynamics in hospitals [17]. This is exemplified convincingly by the success of antibiotic restriction interventions, which lead to a decrease of the resistant strains targeted, [18] (op. cit.) but may result in overuse of replacement drugs and in spreading bacteria resistant to the replacement drugs, i.e., to “squeezing the antibiotic balloon” [19,20,21]. Consequently, understanding how strain dynamics are shaped by the pattern of relationships between drug use and drug resistance is a key information for antimicrobial stewardship efforts aimed at containing antibiotic resistance.

In this particular setting, the resistance spiral was well delineated when examining all major Gram-negative pathogens as one group, but species-by-species analysis revealed differences between them which dynamically evolved over time. These dynamics suggest that the relative importance of a given species may fluctuate, even in the absence of major outbreaks.

The position of *A. baumannii* as the major problem species was stable during the study period, characterized by endemic occurrence of a few carbapenem-resistant clones [22,23]. It persisted as the main species with carbapenem resistance and, consequently, as the main driver of colistin use throughout the five datasets. This is in line with the extensive literature on carbapenem-resistant *A. baumannii*, frequently reported as a major emerging threat among multiresistant nosocomial bacteria. Carbapenem resistance of *A. baumannii* was also linked to extensive use of carbapenems and provoked colistin use in a number of reports [24,25,26].

All other species exhibit changes in relative importance of various consumption–resistance or reciprocal relationships. The role of *P. aeruginosa* seems to be confined to the first two years (2015–2016). Though its cephalosporin was associated with increasing carbapenem use in these two years, this was only weakly linked to carbapenem resistance. Moreover, increasing colistin consumption was associated with decreasing carbapenem resistance. When analyzing the complete resistance spiral, the effect of carbapenem use disappeared and reciprocal associations were the more marked, suggesting that carbapenem resistance of *P. aeruginosa* is less strongly linked to antibiotic consumption than that of *A. baumannii*.

The population structure of *P. aeruginosa* is reported as multiclonal in most studies [27,28]; even when major clones were present, the population was characterized by high diversity and by the presence of several different clones. Similarly, multiple versatile *P. aeruginosa* clones may be circulating in our hospital, as reported in a previous study with a more limited scope conducted in a ward of the same hospital [29]. Apparently, the persistence of one or a few successful clones is unlikely, in contrast to *A. baumannii* [22,23].

*Klebsiella* spp., in contrast to situations in many European countries, were not major players in the resistance spiral in this study. The expected effect of cephalosporin consumption on cephalosporin-resistant *Klebsiella* spp. and its influence on carbapenem use was only detected in the earliest 2015 and latest 2019 datasets. The decreasing importance was foreshadowed in the findings on asymptomatic carriers in the same hospital, where a slow receding was found in parallel with slow emergence of *E. coli* as the major ESBL producing species [30]. Carbapenem resistance has not yet emerged in *Klebsiella* spp. as a major problem in this setting; sporadic occurrence of resistant isolates has been detected, but the incidence is still low. It is tempting to assume a link with the high prevalence of carbapenem-resistant non-fermenters, which may have occupied the niches created by high carbapenem use and may hinder the spread of *Klebsiella*. Increasing use of colistin, which in the 2019 dataset was associated with decreasing carbapenem resistance in *Klebsiella* spp., may also have contributed to this.

In the case of *E. coli*, cephalosporin use was associated with cephalosporin resistance in all datasets but 2019, which, in turn, was the major driver of carbapenem use in 2015 and 2016. This is in line with the emergence of ESBL-producing *E. coli* as one of the main nosocomial threats in our setting [30] and worldwide [31], but, at the same time, suggests its decreasing importance as the driver of carbapenem use. After using carbapenems in fear of multiresistant *E. coli*, carbapenem prescription may have become a prescribing habit or even practice, decoupling from the increasing prevalence of cephalosporin resistance.

The models of the complete spiral supported the above assumption, while in later models carbapenem use seems to be partially substituted by colistin use. This is in line with usage of colistin, i.e., colistin is used primarily when carbapenems are, or are thought to be, ineffective. This was driven primarily by carbapenem resistance of *A. baumannii*, as shown by the steadily observable relationship between carbapenem resistance in *A. baumannii* and colistin use in all datasets. This again points towards the importance of resistance as a provoker of the use of drugs perceived as more efficacious [32]. Colistin resistance was infrequent in the study period, similarly to another University clinic in Hungary [33], and thus was not analyzed.

The presented results lend support to our earlier observation that different species play different roles in the spiral. These roles were dynamic and may change over even as short a time period as one year. Furthermore, these changes in incidence of resistance in a species were linked to changes in antibiotic use, in line with previous findings [14,15]. The present study concentrated on Gram-negative bacteria of major importance, and an important limitation of the study is that Gram-positive bacteria were not investigated, though they are definitely important players in hospital ecology. However, the number of variables which may be used in a single effective model is limited; this shortcoming of the statistical methodology precluded a fully comprehensive analysis incorporating Gram-positive bacteria and the drugs used against them.

We believe that differences detected across the models in this study using the consecutive data series are consequent to the recent changes in the time series compared to the former year’s series, either in the antibiotic consumption or in the resistance series or in their relation. Thus, the present study represents an attempt to track the changes in the resistance spiral over time. The temporally stable dynamics of the Gram-negative resistance spiral supports that antibiotic resistance is shifting towards increasing resistance in Gram-negative bacteria continuously, as extensively documented both at local [8,34,35] and global [36,37,38] scales. This spread of resistance is driven, at least partly, by the turning of the prescribers’ preference towards broader and broader spectrum antibiotics, as concluded earlier [4,8,32].

## 4. Conclusions

In conclusion, the interplay between antibiotic use and resistance in different species was dynamically changing over time. This points out that analyses targeting such relationships provide insight into a momentary situation; assumptions that these relationships are lasting may be misleading. For a more complete and firmer understanding of these relationships and, consequently, to keep stewardship efforts built on these data well-directed, periodic reanalysis of such datasets is advisable.

## 5. Materials and Methods

The study was conducted in a University-affiliated tertiary teaching hospital with 1667 beds between October 2004 and December 2019. We collected monthly consumption data (drugs drawn by wards from the clinical pharmacy) of all cephalosporins, carbapenems, aminoglycosides, quinolones, and colistin in defined daily doses (DDD) per 100 occupied bed-days (OBDs) [39]. Resistance was measured in incidence density of infections per 1000 OBDs by bacteria resistant to cephalosporins, carbapenems, and colistin among *E. coli*, *Klebsiella* spp. (*K. pneumoniae* and *K. oxytoca* together), *P. aeruginosa*, and *A. baumannii* isolated from inpatients. Resistance burden was calculated for all major Gram-negative bacteria by summing up incidence densities of the studied species for each drug group, further referred to as cumulated resistance of Gram-negative bacteria. Multiple isolates from the same patients were included only once.

Five groups of time-series were constituted, the first from October 2004 to December 2015 (dataset 2015), the second from October 2004 to December 2016 (dataset 2016; this dataset is the closest in time to the dataset used in our former study [8]), the third from October 2004 to December 2017 (dataset 2017), the fourth from October 2004 to December 2018 (dataset 2018), and the fifth from October 2004 to December 2019 (dataset 2019).

All five datasets were analyzed using the same modelling strategy as earlier; VAR models were built with all five datasets using the same R script [8]. In unit root determination, lag selection, model building, and model diagnostic, we followed the former protocol, including the rolling window strategy. The association was accepted as existing if it was found in more than 6 of the 12 rolling windows. If the association was significant in only one or two lags, it was termed weak. This rigorously identical modelling strategy allows for robust comparison of results between the five time-series datasets. First, the cumulated resistance of all Gram-negative bacteria was modelled, then models of a single species were built, and, finally, a model capturing the complete resistance spiral was constructed. The models also contained aminoglycoside and quinolone consumption variables in order to represent their effect, but their involvement in the resistance spiral was not tested directly. All corresponding models were compared using the five sets of time-series, i.e., between the five endpoint years.

## Figures and Tables

**Figure 1 antibiotics-10-00734-f001:**
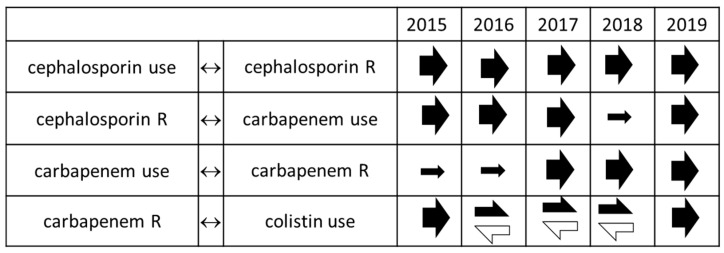
Relationships between antibiotic use and resistance in case of all Gram-negative bacteria together. Filled arrows: positive relationship (effect on response is directly proportionate to the impulse), open arrows: negative relationship (effect on response is inversely proportionate to the impulse), directions of arrows show the direction of the relationship, i.e., arrows pointing right show that the left variable is the impulse and the right variable is the response, arrows pointing left mark the opposite. In this manner, the black-left-white-right arrows mark that carbapenem resistance is associated with increased colistin use, and, at the same time, increasing colistin use is associated with decreasing carbapenem resistance.

**Figure 2 antibiotics-10-00734-f002:**
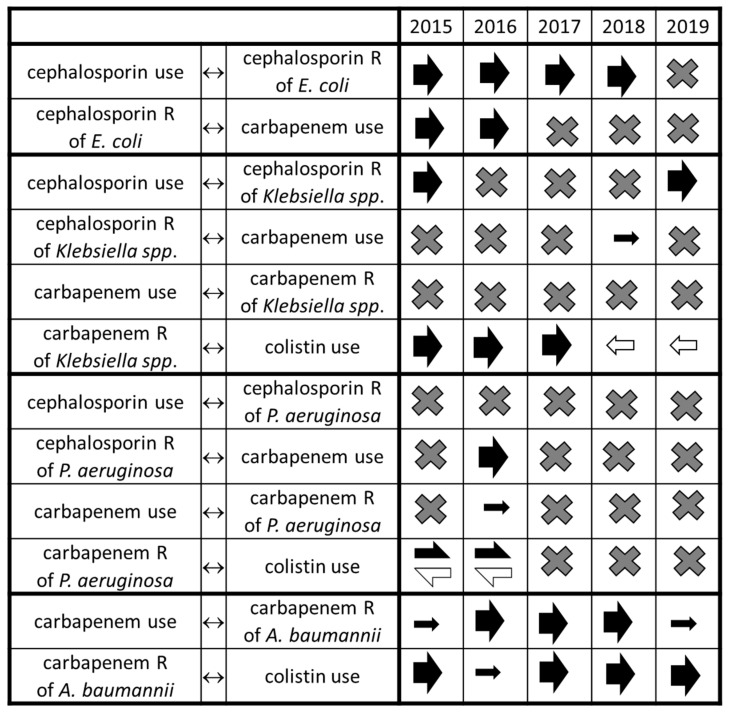
Relationships between antibiotic use and resistance in case of different species analyzed separately. X symbol: no significant relationship; filled arrows: positive relationship (effect on response is directly proportionate to the impulse), open arrows: negative relationship (effect on response is inversely proportionate to the impulse), directions of arrows show the direction of the relationship, i.e., arrows pointing right show that the left variable is the impulse and the right variable is the response, arrows pointing left mark the opposite.

**Figure 3 antibiotics-10-00734-f003:**
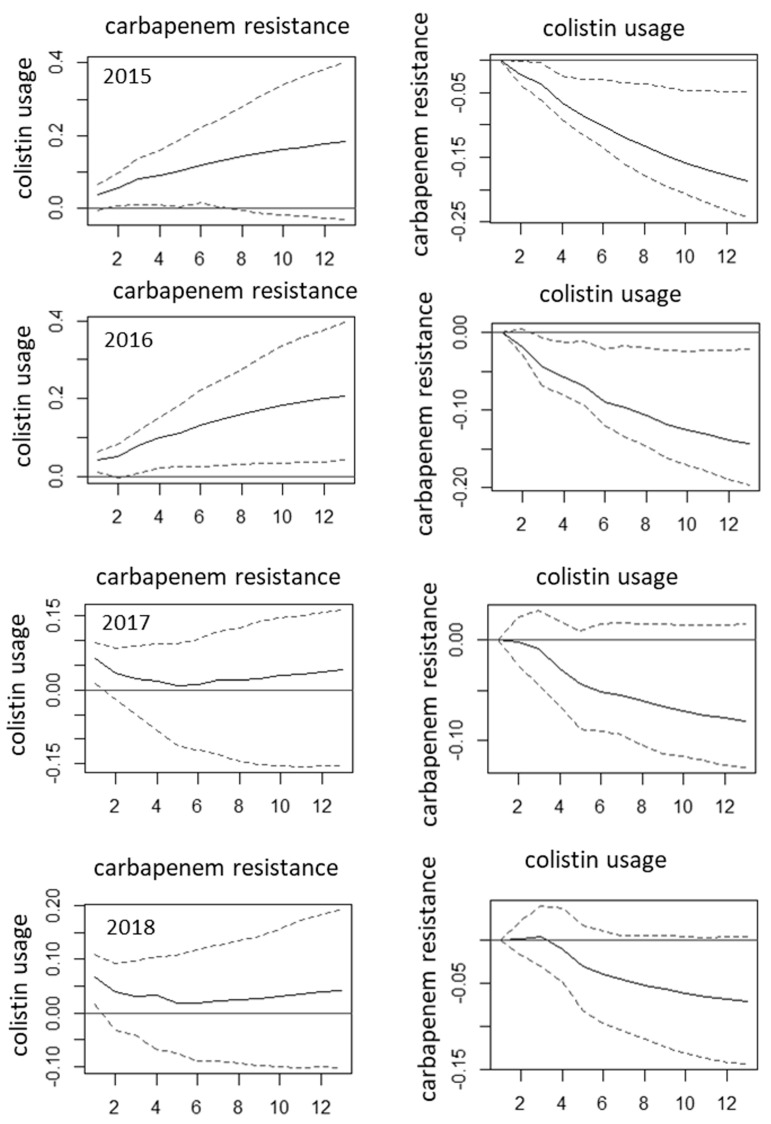
Changes in the relationships between antibiotic use and resistance over time: the example of the relationship between carbapenem resistance in *P. aeruginosa* and colistin use. Left panel: impulse response functions with incidence densities of infections by carbapenem-resistant *P. aeruginosa* per 1000 occupied bed-days as impulses and colistin consumption (defined daily doses per 100 occupied bed-days) as responses. Right panel: impulse functions of the reciprocal relationships with colistin use as impulses and incidence densities of carbapenem-resistant *P. aeruginosa* as responses. Response horizon is shown in the X axis (in months following the impulse), Y axis shows the magnitude of the response. Solid lines are the estimates, dashed lines are the 95% confidence intervals determined by bootstrapping of 100 repetitions. Note that the scales are different. The 2019 results, being similar to 2017 and 2018 results, are not shown due to space constraints.

**Figure 4 antibiotics-10-00734-f004:**
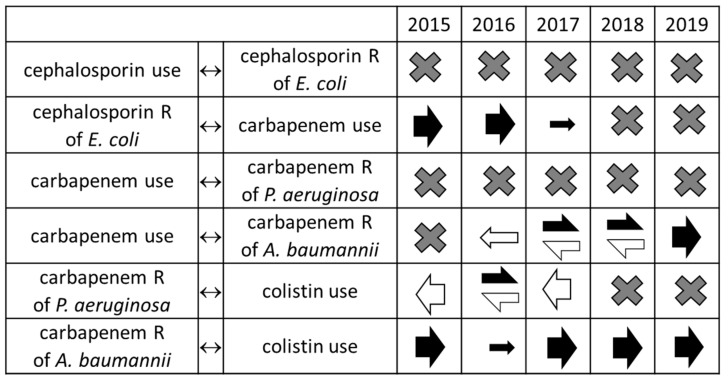
Relationships between antibiotic use and resistance in the model containing the whole resistance spiral. Shaded cells show relationships proved to be unimportant in the species-by-species models and, therefore, not included in the model of the spiral. X symbol: no significant relationship; filled arrows: positive relationship (effect on response is directly proportionate to the impulse), open arrows: negative relationship (effect on response is inversely proportionate to the impulse), directions of arrows show the direction of the relationship, i.e., arrows pointing right show that the left variable is the impulse and the right variable is the response, arrows pointing left mark the opposite.

## Data Availability

Data are available from the corresponding author upon reasonable request.

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
