# Peer review of "Evolution of the Gram-Negative Antibiotic Resistance Spiral over Time: A Time-Series Analysis"

_antibiotics, 2021, doi:10.3390/antibiotics10060734_

Round 1

Reviewer 1 Report

The manuscript : Evolution of the Gram-negative antibiotic resistance spiral over 2 time: a time-series analysis. is a well written work going further into the exploration of the resistance spiral in hospitals. 

Introduction, results and discussion are nicely presented, simply enough to make the message clear. The methods part is quite short but the previous work is accessible with sufficient details. 

Figures are nicely found to put results in a nutshell. 

Figure 3. X axis is not properly defined. I assume it is the number of days ahead of the impulse. 

The authors tackles the question of the resistance spiral for a second time. As previously, it is nicely done, in a comprehensive manner. 

Reviewer 2 Report

Abstract:

…in a tertiary-care hospital in Hungary.

On species-level, association…

Introduction:

…ever since antibiotics were initially introduced to clinical practice. The emergence of resistant bacterial strains initially…

„arms race”

Please discuss the introduction of novel clinical resistance categories, including usual drug resistance (UDR) and difficult-to-treat resistance (DTR), according to the following references:

https://pubmed.ncbi.nlm.nih.gov/30052813/

https://pubmed.ncbi.nlm.nih.gov/32054054/

Please discuss either in the introduction or the discussion, that the increased use of carbapenems may lead to the selection of bacteria in the nosocomial settings, which are intrinsically-resistant to carbapenems (i.e. Stenotrophomonas), which may lead to severe nosocomial infections in immunocompromised patients:

https://pubmed.ncbi.nlm.nih.gov/22232370/

Results:

L72:…across the study years.

L73-74: this sentence is hard to read, please rephrase

L103: appeared as a weak link-how did you quantify this? what was the criteria for the determination for considering the  weak/moderate/strong

The figure captions should be made more concise. In addition, there were some grammatical errors in the descriptions.

Discussion:

L171-172: this sentence needs to be revised for grammar and context. the same goes for the sentence L177-179.

please discuss the inter-relationship between the emergence of carbapenem-colistin resistance and briefly dicuss the epidemiology of colistin resistance in Hungary.

L187: Gram-negative (correct throughout the MS)

A separate and more detailed conclusions section should be included where the authors delineate a main practical message for clinicians/researchers.

Methods:

abbreviated form of bacterial names should be used (as their full names were already described)

The methods used in the study should either be described in more detail (general rule: to allow for anyone else to reproduce this on their own terms!). The r script should be provided as supplementary material

Reviewer 3 Report

The manuscript describes the interplay between antibiotic use and resistance over time in hospitals. The data covers several periods with start time 2004.The applied analysis is interesting and the cited literature is suitable. Such retrospective studies can add valuable information necessary for understanding the impact of the prescription/use of antibiotics on the emergence of resistance. The manuscript can be accepted after a minor revision.

Some minor changes are required:

The authors stated that analysis for fluoroquinolones and aminoglycosides were included in the overall analysis. It is not clear why they did not presented the results for these two group of antibiotics. At least 1-2 sentences are necessary to clarify this issue.

Page 3, Line 96: Please clarify the sentence: "Curiously, cephalosporin resistance was also associated with increasing cephalosporin use ..."
